# A Novel Technique for the Generation of Substantial Numbers of Functional Resident T Cells from Kidney Tissue

**DOI:** 10.3390/cells11142233

**Published:** 2022-07-18

**Authors:** Michiel G. H. Betjes, Frederique Prevoo, Thierry P. P. van den Bosch, Mariska Klepper, Nicolle H. R. Litjens

**Affiliations:** 1Erasmus MC Transplant Institute, Division of Nephrology and Transplantation, Department of Internal Medicine, University Medical Center Rotterdam, P.O. Box 2040, 3000 CA Rotterdam, The Netherlands; f.prevoo@erasmusmc.nl (F.P.); m.klepper@erasmusmc.nl (M.K.); n.litjens@erasmusmc.nl (N.H.R.L.); 2Department of Pathology, Erasmus MC, University Medical Center Rotterdam, P.O. Box 2040, 3000 CA Rotterdam, The Netherlands; t.vandenbosch@erasmusmc.nl

**Keywords:** resident T cells, kidney, biopsy, cell proliferation, interleukin-2, interleukin-15

## Abstract

Studying functionality and antigen-specificity of resident kidney T cells derived from a kidney biopsy is hampered by the lack of sufficient numbers of T cells obtained by the standard method of enzymatic tissue dissociation. Enzymatic dissociation of kidney tissue was compared to a novel method of whole kidney tissue culture allowing T cells to migrate into the medium in the presence of exogenous IL-2 and IL-15. T cell numbers were quantified and phenotype of resident T cells (CD69+CD103+/−), TCR Vβ repertoire and functional characteristics were analyzed with multi-parameter flow cytometry. Renal tissue culture for four weeks in the presence of exogenous IL-2 and IL-15 yielded significantly higher numbers of T cells (1.3 × 10^4^/mm^3^) when compared to cultures without exogenous cytokines (71/mm^3^) or direct isolation by enzymatic dissociation (662/mm^3^ T cells, *p* < 0.05). The proportion of T cells with a resident phenotype did not change in the tissue culture; percentages amounted to 87.2% and 85.1%, respectively. In addition, frequencies of CD4+, CD8+, CD4−CD8−, T cells and MAIT T cells remained similar. For both CD4+ and CD8+, T cells had a more differentiated memory phenotype after tissue culture, but the distribution of TCR Vβ families did not change. In addition, the predominant Th1 cytokine secretion profile and poly-functionality of resident kidney T cell remained intact. T cell proliferation potential was not affected, excluding exhaustion and enrichment of BKV- and CMV-reactive resident T cells was observed. In conclusion, the kidney tissue culture method yields significantly increased numbers of resident T cells without major effects on composition and functionality.

## 1. Introduction

The number of lymphocytes within kidney tissue increases in many pathological conditions causing a local inflammatory response. Several types of kidney lymphocytes have been recognized and associated with acute kidney injury (e.g., double-negative CD3+ T cells) and interstitial fibrosis (natural killer cells or mucosal-associated invariant T (MAIT) cells) [1,2,3,4,5,6]. Recent developments have shown that, similar to other tissues, kidney lymphocytes show expression of CD69 and/or CD103, thereby classifying these cells as resident cells which can be distinguished from migratory, passenger lymphocytes. In kidney transplants, formation of aggregates of lymphocytes develops over time in a substantial number of cases which may progress to formation of structures with characteristics of tertiary lymphoid tissue [7]. These structures can support the local production of antibodies to donor-specific antigens or auto-antibodies and cytotoxic T cells [8,9,10,11]. In kidney transplantation, chronic-active antibody-mediated rejection is the major cause of long-term graft failure, and auto-antibodies facilitate progressive interstitial fibrosis both in native and transplanted kidneys [12,13,14]. The presence of resident T lymphocytes is likely instrumental in this process as these cells support the B cell development into antibody-secreting cells, but their true significance is not known. In addition, areas of interstitial fibrosis frequently contain lymphocytes, but their role, innocent bystander or not, is a matter of debate [15]. To investigate the pathogenic role of kidney T cells in these issues, a larger number of T cells than is usually obtained from a kidney biopsy (<1000 T cells) is needed, in order to perform functional characterization.

The standard method of T cell isolation from kidney tissue is by enzymatic dissociation of kidney tissue cut into small pieces with subsequent isolation of a single cell population. This yields in most cases sufficient numbers of T cells from a relatively large piece of kidney tissue derived from a nephrectomy sample. However, the yield of lymphocytes from a kidney biopsy is usually just enough for limited phenotypical characterization by multi-parameter flow-cytometry [4].

Resident T lymphocytes, e.g., from the skin, may show spontaneous migration into the culture medium during whole tissue culture [16,17]. Clark et al. optimized this protocol by adding IL-2 and IL-15 to the culture medium, which increased the yield of resident T cell numbers without affecting their resident phenotype or T cell receptor (TCR) Vβ-repertoire diversity [18]. This approach has some obvious advantages as the tissue is not tormented by cutting and enzymatic digestion while increased numbers of lymphocytes are harvested untouched.

Analogous to T cell isolation from the skin, a technique of whole kidney tissue culture was developed to maximize the number of isolated kidney T cells which was compared with the results of the standard isolation technique. Detailed phenotypic and functional characterization shows that significant increased numbers of poly-functional, kidney resident T lymphocytes with a polyclonal TCR Vβ repertoire can be obtained by using the novel kidney tissue culture method.

## 2. Materials and Methods

### 2.1. Study Population

Kidney tissue was obtained from non-tumor tissue of donors immediately after nephrectomy for renal cell carcinoma and after transplantectomy of kidney allografts which were removed due to ongoing rejection or loss of function. The volume of the kidney tissue was measured prior to processing. The study was approved by the review board (MEC no: 2020-0704) for nephrectomy and transplantectomy kidney tissue and MEC no: 2019-0213, for kidney allograft biopsies. Table 1 summarizes study population characteristics. All individuals gave written informed consent, and the study was conducted in accordance with the Declaration of Helsinki and Declaration of Istanbul and in compliance with International Conference on Harmonization/Good Clinical Practice regulations.

**Table 1 cells-11-02233-t001:** Clinical characteristics study population.

	Healthy Kidney	Kidney Transplant
	*n* = 10	*n* = 8
Gender (male/female)	5/5	1/7
Age (years), median (IQ range)	68 (57–74)	46 (27–68)
Underlying kidney disease	N.A.	
hypertensive nephropathy		1
diabetic nephropathy		2
polycystic kidney disease		1
SLE-mediated glomerulonephritis		1
granulomatous polyangiitis		1
Nephronophthisis (type 1 homozygous mutation *NPHP*)		1
Unknown		1
Cause of transplant failure	N.A.	
Rejection		5
chronic damage		3
Months since transplantation, median (IQ range)	N.A.	32 (21–52)
Number of immunosuppressive agents at time of sample collection, median (IQ range)	N.A.	2 (1–2)
Donor type (living/deceased)	N.A.	1/7

### 2.2. Isolation of Peripheral Blood Mononuclear Cells

Peripheral blood mononuclear cells (PBMCs) were isolated as described previously [19] and stored at 5 or 10 million/vial at −150 °C until further use.

### 2.3. Enzymatic Dissociation of Kidney Tissue

Kidney tissue was dissociated into single cells using the protocol described by Kildey et al. [4] using mechanic and enzymatic dissociation of kidney tissue with some minor adjustments. Briefly, kidney tissue was cut into small pieces and digestion solution I was added, thereby scaling up the volume accordingly to ensure the pieces of tissue to be digested properly. Digestion solution I contains Hank’s balanced salt solution (HBSS) with calcium and magnesium; ThermoFisher Scientific, Gibco™, Landsmeer, The Netherlands) and Collagenase P (stock 2 mg/mL; Roche Molecular Systems, Woerden, The Netherlands) at a 1:1 ratio. This was incubated for 15 min at 37 °C and halfway as well as at the end, the sample is pipetted up and down several times using a Samco pipette (ThermoFisher Scientific, Gibco™) to mechanically dissociate the cells. Upon a centrifugation step at low speed (600× *g*) at 4 °C for 5 min, the supernatant was harvested and stored. Next, digestion solution II was added, again scaling up the volume used for a kidney biopsy (0.5 mL) and incubated at 37 °C for 10 min. Digestion solution II contains Trypsin-EDTA (stock concentration 0.05%, ThermoFisher Scientific, Gibco™) and HBSS without calcium and magnesium (ThermoFisher Scientific, Gibco™). The solution was pipetted up and down several times at the beginning at end of incubation step using a Samco pipette to mechanically dissociate the cells. After this incubation, an equal amount of culture medium (containing calcium) was added to neutralize trypsin. The kidney tissue should be completely digested with minimal cell clumps present. Sometimes, a white fibrous cord may remain which does not contain cells. This can be easily removed using a Samco pipette. The cell suspension was subsequently centrifuged at 600× *g* at 4 °C for 5 min, and the single cell suspensions were frozen and stored for further analyses.

### 2.4. Explant Cultures of Kidney Tissue

The obtained kidney tissue was split into two parts, one for enzymatic dissociation and the other part for explant culture. We adapted the protocol described by Clark et al. for skin explant cultures [18]. Briefly, the kidney tissue was cut into pieces of 2 mm × 2 mm × 2 mm and 2 pieces were transferred to a well of a 24-well plate (Costar) supplemented with 1 mL of culture medium with or without recombinant human IL-2 (100 U/mL; Alloga BV, Wijchen, The Netherlands) and IL-15 (20 ng/mL; Peprotech, London, UK). Irradiated (40 Gy) autologous PBMCs were added to support lymphocyte survival in the initial experiments but proved to be dispensable. The medium of the explant cultures was refreshed every 2–3 days, and a sample was taken from every well, pooled and evaluated with respect to cell numbers by flow-cytometry once a week (described below). Following a 4-week culture, cells were harvested, frozen and stored for further analyses. In a number of procedures, pieces of kidney tissue before and after culture were processed for immunohistochemistry to evaluate CD3+ T cells within the tissue.

### 2.5. Immunohistochemistry

Six pieces of kidney tissue, 3 derived from nephrectomies and 3 from transplantectomies were histologically evaluated using HE staining and staining for CD3 prior to and following 4 weeks of culture as described before [20,21].

### 2.6. Characterization of T Cell Subsets Obtained from Kidney Tissue

Single cells suspensions were stained for 30 min at room temperature in the dark using antibodies directed to CD3, CD4, CD8 and CD56 following a 15 min staining to exclude dead cells (Appendix A). Kidney tissue resident T cells were defined by their expression of CD69 with or without CD103 [22]. The absolute number of T cells/mm^3^ kidney tissue obtained was determined by flow count beads (Beckman Coulter, Woerden, The Netherlands). Briefly, following staining of cells and upon a wash, the pellet was resuspended in 270 μL FacsFlow (BD, Erembodegem, Belgium). Just before measuring the samples on the flow cytometer, 30 μL FlowCount beads (Beckman Coulter) were added. Absolute cell numbers were calculated as follows:(1)$$\begin{array}{c} \frac{number\;of\;CD3\;+\;events\;\left(cells\right)}{number\;of\;beads\;measured}\;\times \;bead\;concentration\;\left(\frac{beads}{microliter}\right)\left(lot\;specific\right)\;\times \;\\ dilution\;factor\;\left(\frac{30}{300}\right)\;\times \;sample\;volume\;\left(300\;microliter\right) \end{array}$$

To normalize for differences in kidney volume, the absolute number of T cells were divided by the volume of kidney tissue, expressed in mm^3^.

For T cell phenotype analysis, cells were stained with antibodies directed to CD3, CD4, CD8, CD161 (a marker of MAIT cells) and CD69 and CD103 to identify the different T cells (Appendix A). Antibodies directed against CD45RA, CCR7, CD28, and CD27 were included to determine T cell subsets (Appendix A). Naïve T cells were defined as CD45RA+CCR7+, central-memory T cells as CD45RA−CCR7+, effector memory T cells as CD45RA−CCR7− and the highly differentiated Temra cells as CD45RA+CCR7−. An alternative characterization of T cell differentiation status is by CD28 and CD27 expression as both co-stimulatory molecules are expressed by naïve T cells but gradually lost with progressive differentiation to a final highly differentiated T cell status of CD28−CD27− cells [23,24,25,26]. Following extracellular staining, the cells were washed, and intracellular staining was performed using antibodies directed to FOXP3, T-bet and RORγ using the FOXP3/Transcription Factor Staining Buffer according to manufacturer’s instructions (eBioscience, Thermo Fisher Scientific, Bleijswijk, The Netherlands) (Appendix A).

Samples were measured on the BD FACSCanto II or BD Symphony A3 light (BD, Erembodegem, Belgium). Data were analyzed using Kaluza version 2.1 (Beckman Coulter, Woerden, The Netherlands).

### 2.7. Characterization of TCR Vβ-Repertoire of T Cells Obtained from Kidney Tissue

To examine whether clonal expansions of particular TCR Vβ families were induced by explant culture in the presence of IL-2/IL-15, TCR Vβ repertoire analysis was performed on T cells as previously described [27]. Appendix A shows the description of the antibodies directed to the different TCR Vβ-families.

### 2.8. Functional Characterization of T Cells Obtained from Kidney Tissue

Functionality of kidney T cells was assessed by measuring proportions of activation-induced cytokine producing cells and cells expressing the degranulation marker CD107a, the latter a measure of potential cell cytotoxicity of T cells [28]. Cells were thawed and stimulated with a combination of phorbol 12-myristate 13-acetate (PMA; 50 ng/mL; Sigma Aldrich, St Louis, MO, USA) and ionomycin (1 μg/mL; Sigma Aldrich). Stimulation was performed for 15 h in the presence of 5 µL of CD107-a (FITC) (clone H4A3, Biolegend) and the Protein Transport Inhibitor Cocktail (2 μL/mL) (eBioscience, Thermo Fisher Scientific, Bleijswijk, The Netherlands) was included. After stimulation, cells were washed and incubated for 15 min at room temperature with ethylenediaminetetraacetic acid (EDTA) (20 mM; pH 7.2–7.4). Thereafter, the cell surface was stained with antibodies to identify the different T cell subsets (CD3, CD4, CD8, CD161, CD103 and CD69) and after permeabilization, intracellular staining was performed using antibodies directed to interleukin (IL)-2, interferon (IFN)-γ, tumor necrosis factor (TNF)-α, IL-6, IL-17A, and IL-21 (Appendix A). The antibody directed against CD4 was additionally added to the intracellular staining for the PMA/Ionomycin condition, to ensure labeling of internalized CD4 due to stimulation. Resident T cells were identified by CD103 expression in these experiments, as CD69 expression is induced on passenger T cells following PMA/ionomycin stimulation [29,30].

### 2.9. BKV- and CMV-Reactive T Cells and Proliferation Potential of T Cells Obtained from Kidney Tissue

T cells harvested from a 4-week culture of kidney tissue with exogenous IL-2/IL-15 and T cells from PBMCs (both at a concentration of 1–2 × 10^6^/mL) were stimulated with a pool of overlapping peptides (final concentration 1 μg/mL, Miltenyi Biotec B.V., Bergisch Gladbach, Germany) of the immunodominant antigens of CMV (pp65 and IE-1) and BKV (BKV VP1/2, LT and ST) for 18–24 h. Carboxyfluorescein succinimidyl ester (CFSE)-labeled autologous PBMCs at a ratio 1:1 were used as antigen-presenting cells. Antigen-reactive T cells were identified within CFSE-negative cells (responder cells) by CD137-expression [31] and were further characterized using co-expression of CD3, CD4, CD8 and CD69. In addition, antigen-reactive T cells were also evaluated following a 6-day stimulation in the presence of CMV- or BKV peptides and irradiated autologous PBMC, using CFSE-dilution as a measure for proliferation [32]. As a control, cells were left unstimulated or polyclonal stimulated using the T cell mitogen phytohaemagglutinin (PHA, 5 μg/mL, Sigma). Proliferated antigen-reactive T cells were identified as CFSE-negative cells.

### 2.10. Statistical Analyses

Data are depicted as median and interquartile range unless indicated otherwise. Statistical analyses were performed in GraphPad version 9.0 (GraphPad Software Inc., San Diego, CA, USA). A non-parametric Mann-Whitney U test was used for unpaired comparisons, Wilcoxon test for paired comparisons. Correction for multiple testing was done using the Holm-Ŝidák method. *p* < 0.05 was considered statistically significant.

## 3. Results

### 3.1. Lymphocytes Migrate from Kidney Tissue Cultures and Remain a Polyclonal Cell Population

Table 1 summarizes study population characteristics.

After enzymatic digestion, kidney lymphocytes were obtained in variable numbers depending on the source of renal tissue (Table 2).

Transplantectomy yielded on average more T lymphocytes as compared to normal renal tissue obtained after nephrectomy (Appendix A, *p* = 0.02). Translated to a volume of a kidney biopsy, this resulted in 10^4^ − 5 × 10^4^ T cells after standard isolation.

The kinetics of numbers of T cells harvested at different time points after the start of the tissue culture were similar for kidney tissue from a nephrectomy or transplantectomy and followed a log-linear curve (Figure 1A). Standard enzymatic dissociation of kidney tissue resulted in 317 (56–1269) T cells/mm^3^ compared to a median of 1.3 (0.2–4.8) × 10^4^ T cells/mm^3^ after 4 weeks of tissue culture. This corresponded to a median 41-fold (19–384) increase in yield of T cells, although a large variability was observed (Figure 1B). It is noteworthy that low T cell numbers (median 71 T cells/mm^3^) were obtained at 4 weeks of tissue culture without exogenous IL-2 and IL-15.

In some experiments, a tissue sample was obtained from the surgically removed kidney using a 14G biopsy needle. This biopsy-derived kidney tissue was processed in parallel with the remaining kidney tissue, which resulted in similar numbers of T cells per mm3 (data not shown). The TCR Vβ repertoire analysis showed that the T cell population remained polyclonal without evidence of preferred outgrowth of particular Vβ clones (Figure 2), *p* > 0.05).

The kidney tissue before and after 4-week culture were stained for CD3+ T cells. As expected, relatively few T cells were found in the interstitium before culture. After 4 weeks culture, the tubulointerstitial compartment in nephrectomy samples had become largely necrotic while glomerular structures were still recognizable. Remarkably, T cells were now readably detected in the interstitial compartment, indicating that some kidney T cells were retained in the kidney and had proliferated (Appendix A). This is in contrast to the substantial decrease in skin-resident T cells described after a similar time of tissue culture [18].

Thus, using the tissue culture protocol, it was possible to harvest sufficient T cells even from small kidney tissue samples like biopsy material for subsequent detailed phenotypical and functional analysis.

### 3.2. Phenotype of Kidney T Cells after Tissue Culture Remains Largely Unchanged

Kidney T cells were considered resident T cells when expressing CD69 and/or CD103 on their cell surface, markers which are rarely expressed on freshly isolated circulating T cells [29]. Median proportions of resident T cells amounted to 83.8 (74.4–95.2)% of total cells (Table 2). The dominant phenotype of kidney T cells, both CD4+ as well as CD8+, was of the memory phenotype, in particular of the effector-memory subset as judged by CD45RA and CCR7 expression as well as CD28/CD27 expression profile (Table 3). Although the majority of cells was either CD4 or CD8 positive, there was a variable but a substantial % of CD4/CD8 double negative T cells. In line with a recent publication [33], the composition of kidney T cells derived from transplantectomy or nephrectomy tissue was relatively similar with no major statistical significant differences in T cell subpopulations (Appendix A).

After four weeks of tissue culture, the composition of T cells with regard to CD4/CD8 ratio (1.07 before and 1.12 after culture) and the % of CD69+ T cells, with or without co-expression with CD103, was not significantly changed (87.2% and 85.1%, respectively, Figure 1C,D). In the CD69+ and CD69− CD4+ T cells, a relative shift to more differentiation was noted as the average percentage of CCR7−CD45RA+ T cells increased (*p* < 0.01, Figure 3B; from 2.3% to 40.6% and from 4.6% to 40.3%, respectively), which was not evident by CD27/CD28 expression (Figure 3C). For CD8+ T cells, in particular, the fraction of CD28−CD27− CD69+ T cells increased (*p* < 0.01), which is indicative of relatively more differentiated cells, but in contrast to the CD4+ T cells, the differentiation status based on CCR7/CD45RA expression remained relatively similar. It is worth mentioning that the relative small percentage of T cells with a naïve phenotype remained unchanged after four weeks culture.

### 3.3. Profiles of Cytokine- and CD107a—Expressing Kidney T Cells before and after Tissue Culture

The profile of cytokine-expressing kidney T cells was dominated by IFN-γ, TNF-α and IL-2, with low frequencies of IL-6-, IL-17A- and IL-21-producing cells, consistent with a dominant Th1 signature of kidney T cells. Resident CD103+CD4+ T cells had higher (*p* < 0.01) proportions of TNF-α, IL-2, IFN-γa, IL-17A and IL-21-producing cells compared to CD103− CD4+ T cells (Figure 4B,C). Resident CD103+CD8+ T cells had higher proportions of IFN-γ (*p* = 0.04), IL-6 (*p* < 0.01), IL-17A (*p* < 0.01) and IL-21 (*p* = 0.02) producing cells compared to CD103− CD8+ T cells (Figure 4E,F). After 4 weeks culture, the cytokine expression profiles remained largely unchanged except for an increase in the % of CD107a+ CD4+ (from a median 22% to 55%, *p* < 0.01), CD107a+ CD8+ T cells (from a median 38% to 89%, *p* < 0.01) and an increase in % TNF-ᾳ+ CD4+ CD103− T cells (from a median of 67% to 94%, *p* < 0.01). These changes were seen for kidney T cells from both transplantectomy and nephrectomy derived kidney tissue cultures (Figure 4).

However, the CD107a expression of T cells directly isolated from transplantectomy tissue was higher compared to nephrectomy-derived T cells while the % of inflammatory cytokine producing cells tended to be lower (Appendix A). Poly-functionality of T cells was assessed calculating the number of factors (two or more cytokines and/or CD107a, a marker for degranulation) expressed per cell. The proportion of cells expressing 4 factors/cell was significantly increased compared to direct isolated kidney T cells for both CD4+ resident (from a median 3% to 18%, *p* = 0.01, Figure 5B) and passenger T cells (from a median 3% to 14%, *p* < 0.01, Figure 5C).

Expression of transcription factors showed a predominant Tbet-positive signature, in line with the Th1 cytokine expression profile of the kidney T cells. T cells harvested after tissue culture showed more Tbet-positivity and an increase of RORγ-expressing T cells (Figure 6).

### 3.4. CMV- and BKV-Reactive Kidney T Cells before and after Culture

Kidney resident T cells may harbor specific antigen-reactive T cells against CMV and BKV [5]. In particular, the latter is an example of enrichment for specific resident T cells that exert tissue-specific anti-viral responses like for instance influenza-specific resident T cells in the lung parenchyma [8]. Kidney T cells obtained after culture were specifically tested for the frequency of BKV and CMV-reactive T cells, which showed an increased frequency of CMV-and BKV-reactive resident kidney T cells compared to the peripheral blood PBMC (Figure 7). For BKV-reactive (i.e., CD137+) T cells, this was observed in both the CD4+ (median 1.35% vs 0.16%, *p* = 0.03) and CD8+ T cell population (median 0.50% vs 0.0%). In accordance with this, both antigen-specific CD4+ and CD8+ T cell proliferation was observed, which was substantially more for BKV-reactive kidney T cells compared to the peripheral blood T cells. Polyclonal lectin-induced T cell stimulation with PHA caused vigorous proliferation of kidney tissue culture-derived T cells, definitively excluding a possible state of exhaustion of kidney T cells after culture.

## 4. Discussion

The results of the present study show that kidney tissue cultures can yield substantial more T cells per mm3 tissue as compared to the standard method of enzymatic tissue digestion. Already some decades ago, several studies attempted to isolate T cells by tissue culture of transplanted kidney samples under different experimental conditions [34,35,36]. Although enough T cells from rejection kidneys could be harvested for functional analysis, kidney tissue cultures from non-rejecting [36] or kidneys prior to transplantation [35] yielded no T cells. Using the current protocol, the increased yield of T cells even from a kidney biopsy of a native non-inflamed kidney allows for performing multi-parameter phenotypical and functional analysis. The fold expansion of T cells was higher compared to the 10-fold expansion reported for the original skin-tissue culture protocol [18].

In accordance with other studies on kidney T cells (and other tissue-resident T cells), the T cells are predominantly of the effector-memory phenotype, secrete multiple cytokines, and CD4+ T cells are predominantly of the Th1 subtype. Expression of CD69 with or without co-expression of CD103 is the defining characteristic of resident T cells (reviewed in [1]), features that remained remarkably stable after tissue culture.

Prolonged in vitro culture periods with T cell survival and lymphocyte growth cytokines like IL-2 and IL-15 may give rise to unwanted outgrowth of particular cell types. However, the ratio of CD4+ and CD8+ T cells remained similar and the percentage of cells with a naïve phenotype did not decline, indicating homeostatic proliferation and not differentiation of these T cells. The CD4+ T cell differentiation status showed a relative increase in highly differentiated EMRA T cells based on CCR7/CD45RA expression patterns, while in the CD8+ T cell population a relatively modest shift towards more highly differentiated memory T cells was observed based on the loss of both CD27 and CD28 cell surface expression. After culture, the stimulation-induced CD107a and Th1-type signature transcription factor Tbet expression increased, indicating a higher cytotoxic potential of Th1 type T cells, which fits with the relative increase of differentiated effector-memory T cells. Importantly, the TCR Vβ repertoire was not affected by the culture period, and no indication was found for dominant outgrowth of particular T cell clones. This result is in accordance with what was reported about the resident skin T cells obtained after whole skin tissue cultures [18].

The functional capacities of the T cells as defined by proportion of cytokine producing cells remained intact as was the proportion of poly-functional T cells, i.e., those able to produce two or more different cytokines and/or expressing the marker for degranulation, CD107a. These results indicate that the T cells are not becoming senescent/exhausted and still produce cytokines upon stimulation. In accordance with these findings, polyclonal lectin-induced T cell proliferation remained vigorous in the kidney tissue-culture derived T cells. In order to test their functionality with a more physiological stimulus, the presence and proliferation of CMV- and BKV-reactive T cells was studied. The results not only showed that cells remain responsive to these antigens but also that kidney resident T cells have a higher frequency of the antigen-reactive T cells as compared to the circulation. In particular, BKV-reactive T cells are more abundantly present in the kidney. This finding confirms the recent results published by Dornieden et al., who used dextramers to identify CMV- and BKV-specific kidney tissue-isolated CD8 T cells [5].

The advantage of the kidney tissue culture method is that in general enough T cells can be obtained for both CD4+ and CD8+ T cell analysis with respect to antigen-specificity and proliferative response. Current single cell techniques involving single cell RNA sequencing have shown to be a powerful analytic potential, e.g., showing the landscape of immune cells present in the kidney at high granularity [37]. However, for translational immunological research questions like the search for which kidney T cell type is pathogenic, analysis of the functional status and the antigen-specificity of the isolated T cells is highly instrumental. However, it should be realized that there is a shift towards more differentiated memory T cells which can limit the interpretation of results, depending on the type of experiments performed.

A detailed comparison between kidney T cells from normal kidney tissue and a failing kidney graft was not the objective of the study, but the data obtained are worthwhile to consider. The number of T cells obtained per mm^3^ was significantly higher in transplantectomy tissue (which is not unsurprising given the inflammatory conditions of these kidneys) [38], but the differences (before and after culture) in T cell subset composition, cytokine expression profiles, transcription factors and tissue culture characteristics were relatively minimal to modest. This unexpected finding was also reported in a recent study by van der Putten et al. [33] and indicates that other data are needed, e.g., determination of antigen-specificity, studies on control mechanisms of effector functions or T cell metabolic state [39], to assess whether the tissue-resident T cells are active and deleterious or in a resting state.

In conclusion, the newly developed kidney tissue-culture protocol allows for recruitment of high numbers of functional resident kidney T cells facilitating functional analysis.

## Figures and Tables

**Figure 1 cells-11-02233-f001:**
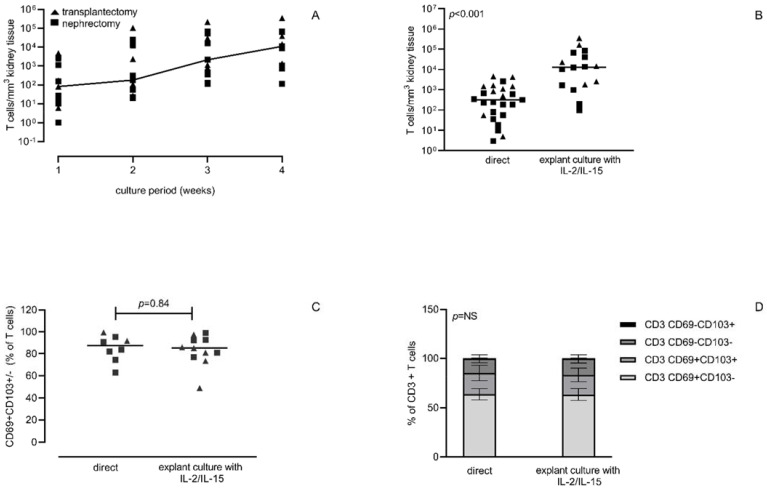
Increased T cell numbers following explant culture of different sources of kidney tissue with exogenous IL-2 and IL-15. T cell numbers (*y*-axis) were determined on a weekly basis during four weeks of explant culture in the presence of exogenous IL-2 and IL-15 for the different sources of kidney tissue (*n* = 10 nephrectomies and *n* = 6 transplantectomies), using flow cytometry thereby normalizing for the different volumes of kidney tissue obtained (**A**). data depicted representing the median of the different samples per week. In (**B**), the final number of T cells is depicted on the *y*-axis, again normalizing for the amount of kidney tissue obtained. In (**C**), the percentage of CD69+CD103+/− T cells is depicted. The horizontal line represents the median of the different samples. The squares represent data obtained for nephrectomies and the triangles’ display data obtained for transplantectomies. In (**D**), the distribution of CD69 and CD103 for CD3+ T cells is depicted (average ± SEM).

**Figure 2 cells-11-02233-f002:**
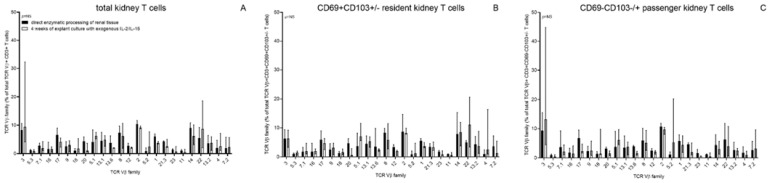
TCR Vβ repertoire T cells. The TCR Vβ repertoire was determined using the flow cytometry-based IOTest^®^ Beta Mark TCR Vbeta repertoire kit for T cells obtained following direct enzymatic dissociation of kidney tissue (closed bars) and compared to that of T cells harvested following four weeks of explant culture in the presence of exogenous IL-2 and IL-15 (open bars). Kidney tissue used was obtained from three nephrectomies and one transplantectomy. In (**A**), the repertoire for total kidney CD3+T cells is depicted, in (**B**), that of CD69+CD103+/− resident kidney CD3+ T cells, whereas (**C**) depicts the CD69− CD103+/− passenger kidney CD3+ T cells. Bars represent median (IQ range) values for the 24 different TCR Vβ families expressed as a percentage of total TCR Vβ+ cells.

**Figure 3 cells-11-02233-f003:**
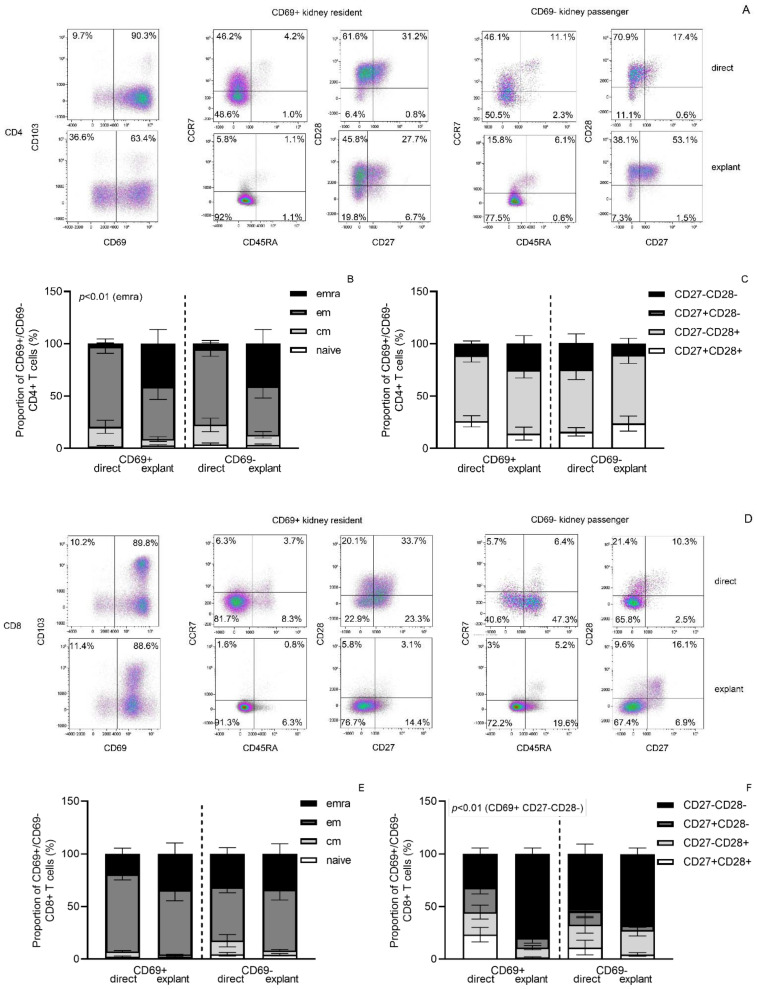
Explant culture in the presence of exogenous IL-2 and IL-15 resulted in more differentiated memory T cell subsets compared to T cells obtained following direct enzymatic processing of kidney tissue. In (**A**,**D**) a typical example of the flow cytometric gating strategy for phenotype of CD4+ (**A**) and CD8+ (**D**) T cells is depicted for those obtained following direct enzymatic processing of kidney tissue (**first row**) and those upon four weeks of explant culture in the presence of exogenous IL-2 and IL-15 (**second row**). T cells were characterized by flow cytometry following staining for T cell subsets using either antibodies directed to CCR7 and CD45RA (**B**,**E**) or CD28 and CD27 (**C**,**F**). Average (±SEM) proportions of T cells obtained from *n* = 11 kidney tissues (*n* = 5 nephrectomies and *n* = 6 transplantectomies) are depicted in stacked bars. CM = central memory, EM = effector memory, EMRA = CD45RA+ terminally differentiated effect or memory T cells. Naïve T cells are identified as CCR7+CD45RA+, CM T cells as CCR7+CD45RA−, EM as CCR7−CD45RA− and EMRA as CCR7−CD45RA+.

**Figure 4 cells-11-02233-f004:**
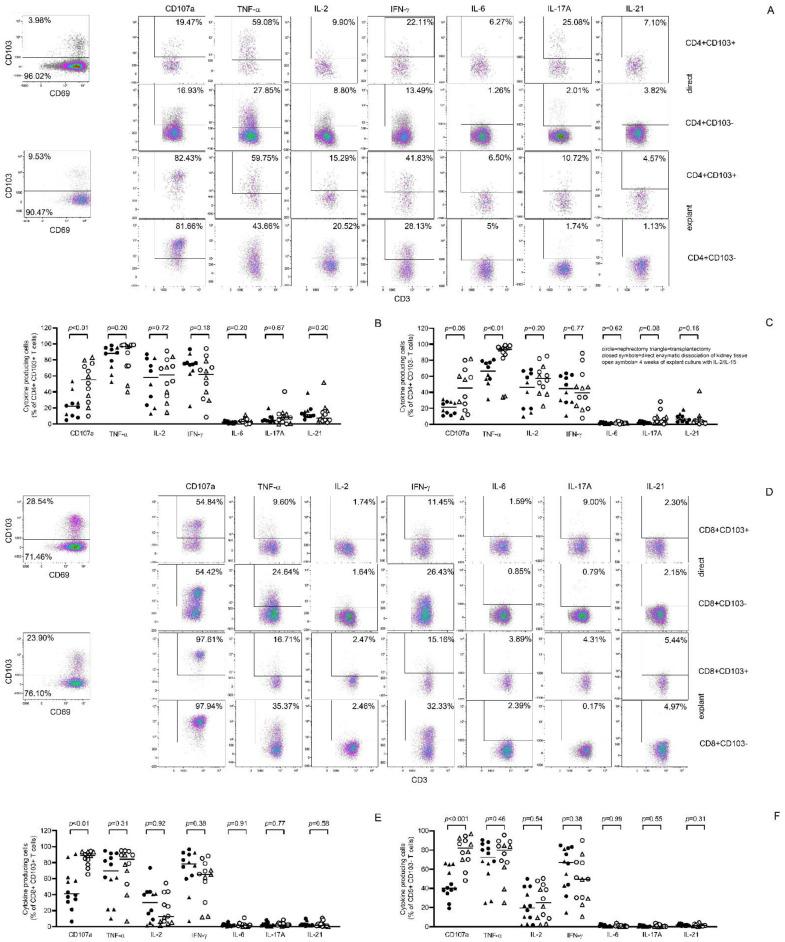
Pro-inflammatory profile was maintained following 4 weeks of explant culture in the presence of exogenous IL-2 and IL-15. Proportions of pro-inflammatory cytokine producing and CD107a- expressing cells were determined upon overnight stimulation of T cells with PMA and ionomycin. In (**A**) (CD4+) and (**D**) (CD8+), a typical flow cytometric example of the gating strategy is depicted. In the upper panel, T cells obtained direct following enzymatic dissociation are depicted (CD103+ in first row, CD103− in second row), whereas in the lower panel that of T cells harvested following 4 weeks culture kidney tissue is given. Percentages of cytokine producing cells are depicted for resident (CD103+) as well as passenger (CD103−) kidney CD4+ (**B**,**C**) and CD8+ (**E**,**F**) T cells either upon direct enzymatic processing of kidney tissue (closed symbols) or following four weeks of explant culture in the presence of exogenous IL-2 and IL-15 (open symbols). Kidney tissue was obtained from five transplantectomies (triangles) and seven nephrectomies (circles).

**Figure 5 cells-11-02233-f005:**
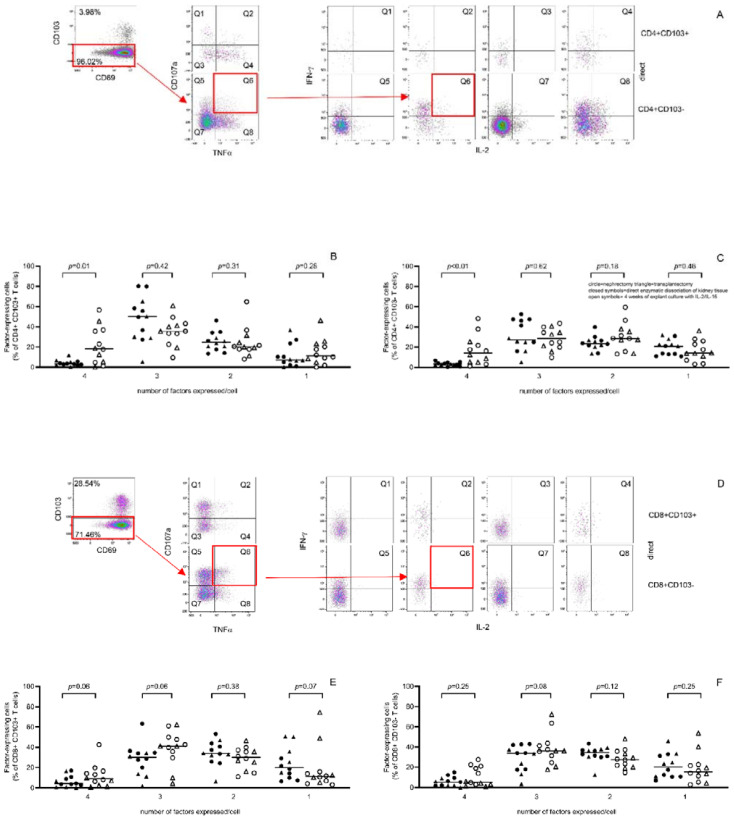
Poly-functional nature of T cells was maintained upon four weeks of explant culture in the presence of exogenous IL-2 and IL-15. A typical example of the flow cytometric gating strategy to evaluate the poly-functional nature of T cells is given in (**A**) (CD4+ T cells) and (**D**) (CD8+ T cells). For example, CD103− T cells, highlighted by a red rectangle, are depicted in a dotplot with TNF-α on the *x*-axis and CD107a on the *y*-axis, cells positive for both (second red rectangle, Q6) are depicted in a dotplot, depicting IL-2 on the *x*-axis and IFN-γ on the *y*-axis and cells positive for both here represent cells expressing all factors examined (third red rectangle). The poly-functional nature (i.e., either producing 4,3, 2 or 1) of the pro-inflammatory Th1-cytokines TNF-α, IL-2 and IFN-γ producing and/or CD107a-expressing cells was determined for resident (CD103+) as well as passenger (CD103−) CD4+ (**B**,**C**) and CD8+ (**E**,**F**) T cells either upon direct enzymatic processing of kidney tissue (closed symbols) or following four weeks of explant culture in the presence of exogenous IL-2 and IL-15 (open symbols). Kidney tissue was obtained from five transplantectomies (triangles) and seven nephrectomies (circles).

**Figure 6 cells-11-02233-f006:**
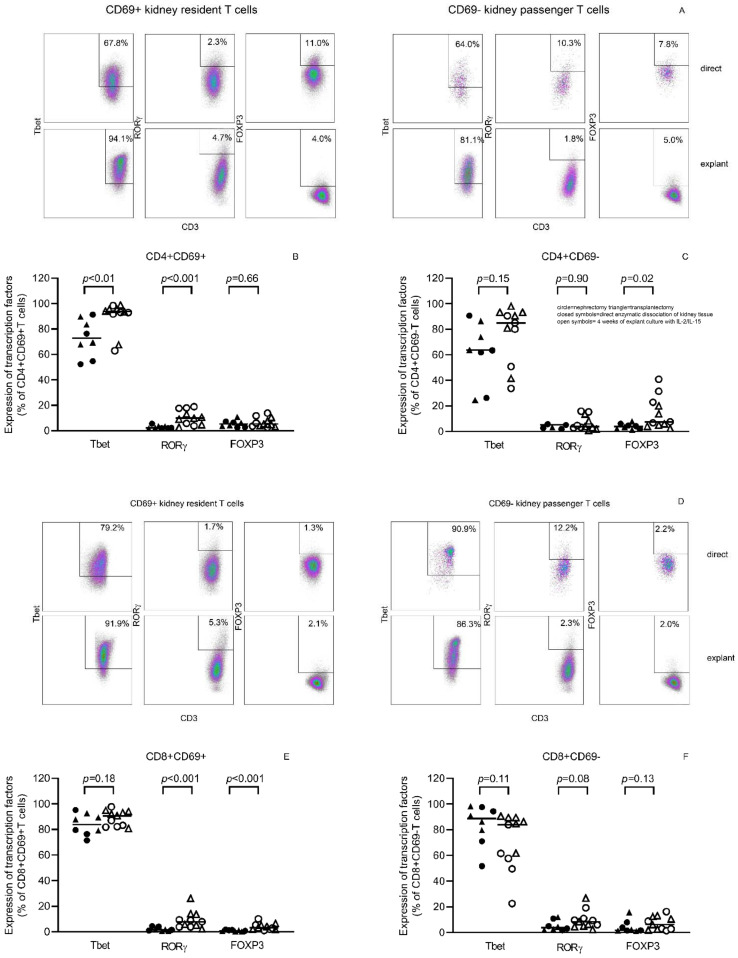
T cells harvested from explant culture in the presence of exogenous IL-2 and IL-15 showed increased expression of the Th1 transcription factor Tbet compared to T cells obtained following direct enzymatic processing of kidney tissue. In (**A**,**D**), a typical example of the flow cytometric gating strategy for analysis of expression of transcription factors of CD4+ (**A**) and CD8+ (**D**) T cells is depicted for those obtained following direct enzymatic processing of kidney tissue (**first row**) and those upon four weeks of explant culture in the presence of exogenous IL-2 and IL-15 (**second row**). T cells were characterized by flow-cytometry following staining for different transcription factors, i.e., Tbet, RORγ and FOXP3. Proportions of transcription factor expressing CD69+ and CD69− CD4+ (**B**,**C**) and CD8+ (**E**,**F**) T cells obtained from *n* = 11 kidney tissues (*n* = 5 nephrectomies, circles and *n* = 6 transplantectomies triangles) are depicted. The horizontal line represents the median.

**Figure 7 cells-11-02233-f007:**
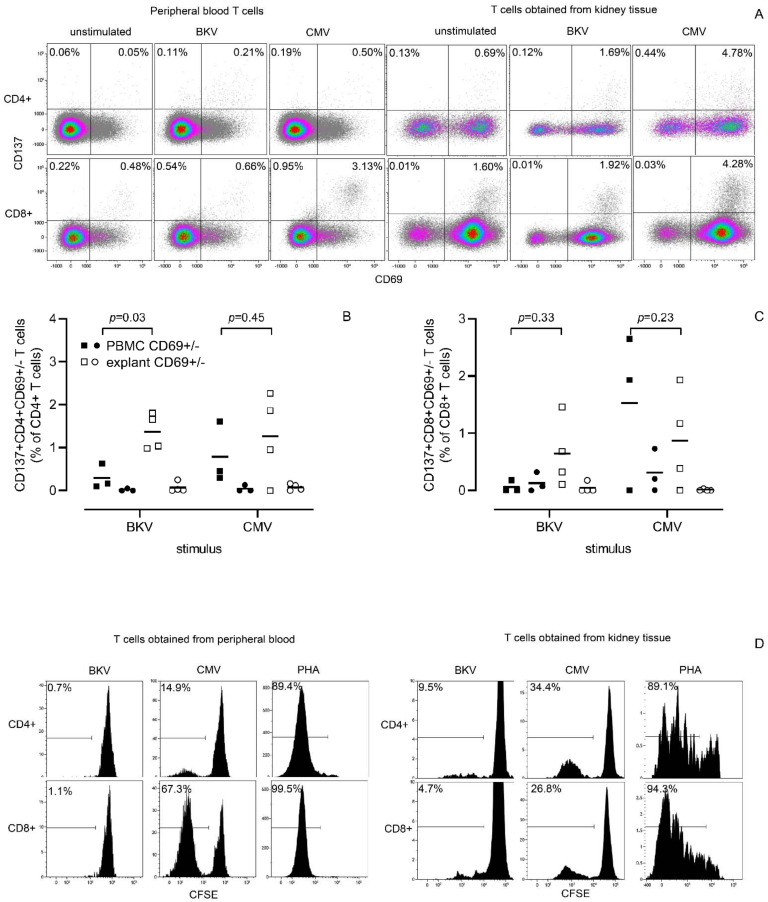
T cells harvested following a 4-week kidney tissue culture in the presence of exogenous IL-2 and IL-15 are functional. Functionality of T cells harvested following a 4-week culture of kidney tissue in the presence of IL-2 and IL-15 (**right panel**) was compared to that of T cells within peripheral blood mononuclear cells (**left panel**), using a short-term stimulation and CD137-assay as read-out of activation (**A**) as well as a 6-day stimulation using dilution CFSE as a readout of proliferation (**D**). In A, a typical example is depicted for the different stimuli depicting CD69 on the *x*-axis as CD137 on the *y*-axis. Percentages depicted are % CD137-expressing T cells within those lacking or co-expressing CD69, respectively. Numbers depicted are CD137+ T cells within those lacking or co-expressing CD69 expressed as a percentage of CD4+ (**B**) and CD8+ (**C**) T cells with those from PBMC in closed and those from explant in open symbols, respectively. Squares represent CD69+ T cells and circles that of CD69- T cells. Data, corrected for background, are from four individuals, two of them underwent a transplantectomy and two a nephrectomy, from which PBMCs were also available. The horizontal line represents the median. In (**D**), a typical flow-cytometric example for analysis of proliferation is given with dilution of CFSE depicted on the *x*-axis and amount of events on the *y*-axis. Percentages depicted are % CFSE- CD4+ (**top**) and CD8+ (**bottom**) T cells, respectively. The closed histogram represents T cells stimulated with either BKV, CMV or PHA. T cells within peripheral blood are depicted on the left side and on the right side those harvested following four weeks of explant culture in the presence of IL-2/IL-15. Peripheral blood T cells and kidney tissue are from the same individual, i.e., a CMV-seropositive kidney transplant recipient that has developed BKV nephropathy following transplantation. The renal tissue was obtained from a transplantectomy. Stimuli include overlapping peptide pools of BKV and CMV. The negative and positive controls include cells left unstimulated and stimulated with PHA, respectively (the latter only for the proliferation assay).

**Table 2 cells-11-02233-t002:** Lymphocyte populations following direct enzymatic dissociation of kidney tissue.

Median and IQ Range *n* = 8 Kidney Tissues	Lymphocyte Population	Resident (CD69^+^CD103^+/−^) (%)	Passenger (CD69^−^CD103^−/+^) (%)
T cells (cells/mm^3^) median	317 (56–1269)	87.24 (76.28–94.30)	12.77 (5.71–23.71)
CD4^+^T cells (% of CD3^+^ T cells)	47.71 (26.68–59.16)	89.64 (76.64–94.64)	10.37 (5.37–23.36)
CD8^+^T cells (% of CD3^+^ T cells)	45.55 (31.40–65.00)	88.45 (81.38–95.55)	11.56 (4.44–18.63)
CD4^−^CD8^−^T cells (% of CD3^+^ T cells)	4.98 (3.06–7.02)	81.29 (73.76–89.4)	18.72 (10.61–26.25)
CD161^+^T cells (% of CD3^+^ T cells)	14.55 (10.35–17.25)	95.13 (91.73–98.7)	4.88 (1.30–8.27)
NK cells (cells/mm^3^)	12 (8–81)	76.84 (64.01–90.53)	23.17 (9.47–35.99)

**Table 3 cells-11-02233-t003:** CD4+ and CD8+ T cell phenotype characteristics following direct enzymatic dissociation of kidney tissue.

	Resident (CD69^+^CD103^+/−^) (%)	Passenger (CD69^−^CD103^−/+^) (%)	*p* **
CD4^+^T cell subsets			
Naive (CD45RA+CCR7+) *	1.54 (0.23–3.46)	3.72 (0.33–6.71)	0.02
CM (CD45RA−CCR7+) *	9.06 (6.70–36.43)	8.76 (6.07–40.52)	0.84
EM (CD45RA−CCR7−) *	89.14 (55.99–90.07)	85.04 (47.16–89.27)	<0.01
EMRA (CD45RA+CCR7−) *	1.99 (1.13–3.41)	4.14 (2.56–6.93)	<0.01
CD27+CD28+ *	30.18 (9.35–33.23)	14.30 (4.47–26.07)	<0.01
CD27−CD28+ *	58.11 (50.12–82.09)	63.32 (34.30–85.07)	1.00
CD27+CD28− *	0.78 (0.15–1.39)	0.67 (0.15–1.63)	0.84
CD27−CD28− *	7.42 (5.09–15.67)	12.05 (7.08–40.22)	0.02
Tregs (CD45RA+/−FOXP3+) *	2.06 (0.91–2.42)	1.39 (0.72–2.66)	0.83
Tbet+	72.86 (57.93–88.21)	63.76 (35.21–83.44)	0.08
RORgamma+	2.21 (1.89–3.72)	5.22 (2.91–9.14)	0.02
FOXP3+	5.34 (2.91–9.14)	3.87 (1.89–5.81)	0.04
CD8^+^T cell subsets			
Naive (CD45RA+CCR7+) *	1.65 (0.81–3.54)	3.78 (0.83–6.32)	0.11
CM (CD45RA−CCR7+) *	4.31 (3.56–6.82)	5.32 (2.70–17.12)	0.38
EM (CD45RA−CCR7−) *	78.70 (60.88–83.88)	45.92 (37.16–69.45)	<0.01
EMRA (CD45RA+CCR7−) *	14.54 (7.44–31.61)	24.93 (15.98–50.04)	0.05
CD27+CD28+ *	18.69 (5.10–38.65)	3.54 (0.83–13.51)	<0.01
CD27−CD28+ *	14.85 (5.92–42.66)	17.12 (5.81–30.68)	0.84
CD27+CD28− *	21.59 (4.43–43.67)	7.84 (0.38–17.17)	0.08
CD27−CD28− *	30.62 (13.19–48.10)	58.15 (31.55–75.33)	0.04
Tbet+	83.75 (77.12–91.85)	88.61 (73.32–96.86)	0.90
RORγ+	1.57 (0.70–3.51)	3.83 (2.34–9.45)	<0.01
FOXP3+	0.97 (0.33–1.41)	1.95 (1.46–6.93)	<0.01

Median and IQ range depicted for eight kidney tissue samples; * proportions within CD69+CD103+/− (resident) and CD69−CD103− (passenger) T cells, respectively; ** CD69+ versus CD69− populations were compared using a Wilcoxon matched pairs signed rank test.

## Data Availability

Data are contained within the article or Appendix A.

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
