# Peer review of "A Novel Technique for the Generation of Substantial Numbers of Functional Resident T Cells from Kidney Tissue"

_cells, 2022, doi:10.3390/cells11142233_

Round 1

Reviewer 1 Report

In this study, Betjes et al. report an original protocol for the study of kidney-resident T cells. Such cells are rare in small kidney biopsies, and the use of this novel technique is of great interest for functional and phenotypic studies of T-cell populations. The paper is well written and, method is sound and the conclusions are largely supported by experimental data. Nevertheless, a few points should be addressed:

Major points:

-Error bars, and more importantly, statistical analysis, are missing in figures 2 and 3, preventing and conclusive analysis.

-The methods for enzymatic dissociation of kidney tissue and FACS staining should be detailed in full, as they are key protocols in the paper.

-The viral specificity of expanded T cells (Figure 7) is of critical importance for further functional analyses. Therefore, it would be insightful to show (is available/feasible) the proportion of virus-specific T cells BEFORE the expansion (or at earlier times of culture). .

Minor points

-Line 86 : -150 Co should be -150°C

-Lines 251-253 : This might be an overstatement, as no statistical analysis is shown. For instance the abundance of Vb22+ cells seems to increase.  

-Results, section 3.3 : The % of TNF+ CD4+CD103- is significantly increased, this should be specified in the text.

-The resolution of most figures is too low. This should be drastically improved before publication.

-Figure legends are somehow complicated/too long and could be simplified.  

-In figure 1, the correspondence of symbols (showing nephrectomy vs transplantectomy) should be displayed on the figure. Same comment in figure 4-5-6 for before/after culture symbols.

Reviewer 2 Report

Betjes et al. developed a novel technique for the generation of substantial numbers of 2 functional resident T cells from kidney tissue. The study is interesting and novelty. The manuscript is written but requires additional editorial input. The methods and results are well controlled. The reviewer has several minor concerns.

1. The authors should analyze the numbers of 2 functional resident T cells in healthy kidneys and kidneys with kidney diseases in animal models using enzymatic dissociation of kidney tissue and the novel technique in this MS, respectively.

2. Please delete L179-L181.
